# On-Demand Dynamic Terahertz Polarization Manipulation Based on Pneumatically Actuated Metamaterial

**DOI:** 10.3390/mi14112094

**Published:** 2023-11-12

**Authors:** Yongchao Zou, Yan Wang, Yangjian Zeng, Pan Xu, Zhengliang Hu, Hongbin Yu

**Affiliations:** 1College of Meteorology and Oceanography, National University of Defense Technology, Changsha 410073, China; zouyongchao@nudt.edu.cn (Y.Z.);; 2School of Optical and Electronic Information, Huazhong University of Science and Technology, Wuhan 430074, Chinazyj547955107@163.com (Y.Z.); 3Optics Valley Laboratory, Wuhan 430074, China; 4Wuhan National Laboratory for Optoelectronics, Huazhong University of Science and Technology, Wuhan 430074, China

**Keywords:** terahertz metamaterial, dynamic modulation, polarization manipulation, pneumatic actuation

## Abstract

In this paper, a new tuning strategy is proposed by incorporating a pneumatically actuated metamaterial to achieve on-demand polarization manipulation at THz frequencies. Through controlling the actuation pressure, the device function can be flexibly switched among three types of polarization conversion capabilities within the same operation frequency band, from 1.3 THz to 1.5 THz, in which the mutual conversion between linear polarization and circular polarization, such as a quarter-wave plate, and handedness inversion between circular polarizations as a helicity inverter as well as a helicity keeper, have been successfully achieved between the incidence and reflection. Moreover, the intrinsic tuning mechanism for the polarization manipulation is also discussed.

## 1. Introduction

Polarization is one of the most important characteristics of the electromagnetic (EM) wave; it has been widely used in various applications, such as imaging, detection and optical communication [1,2,3]. The polarization of the EM wave can be divided into three types, linearly polarized (LP) waves, circularly polarized (CP) waves and elliptically polarized (EP) waves, and the manipulation of the polarization states is highly useful and desirable. However, conventional polarization converters, which work based on the birefringence effect, dichroic crystal and optical grating [4,5,6], suffer from several issues, such as a bulky size, low efficiency, a narrow bandwidth and difficult integration into compact systems. As a result, the exploration of compact and lightweight polarization converters is still ongoing.

Metamaterials, as artificial materials constructed by subwavelength microstructures, have been attracting more and more research interest due to their diverse electromagnetic properties that are unavailable for natural materials [7]. Such unique properties provide an alternative approach to manipulating the EM wave polarization. To date, great efforts have been made to design polarization control devices based on metamaterials [8,9,10,11,12]. Nevertheless, the conventional polarization-manipulated meta-devices are of the passive type, and their functionalities are predefined by their design and remain unchanged once they are fabricated, thus largely limiting their application adaptability and flexibility. From a real application point of view, it will be much more desirable to provide controllable polarization manipulation capabilities with the same meta-device. As a result, in recent years, researchers have focused on the development of active/tunable metamaterials and various new technologies have been successfully demonstrated [13,14,15,16,17,18]. As for the dynamic polarization control in the THz range, a strategy based on the adoption of functional materials has been widely reported due to its fast response advantage. In Refs. [19,20,21,22,23], different polarization conversion functions, such as linear polarization converters and linear-to-circular polarization converters, are obtained through electrical gating and/or the doping of graphene. At the same time, as a three-dimensional analogue of graphene, the Dirac semimetal (DSM) is also proposed to achieve multiple functions in a single metamaterial structure, by dynamically changing the Fermi energy and relative permittivity of DSM with external electric stimulation [24,25]. In Refs. [26,27,28,29,30], the phase transition characteristic between the metallic and insulating states of vanadium dioxide is used to construct different polarization manipulation functions. More recently, a new tunable material called Ge2Sb2Te5 (GST) has been developed to build reconfigurable polarization conversion functions due to its different optical properties in the amorphous and crystalline states [31,32]. Moreover, to improve the performance and enhance the function further, a novel strategy of combining different functional materials together has also been designed, as reported in Refs. [33,34,35,36]. Although great achievements have been realized, the involvement of functional materials as well as their specialized fabrication requirements still bring issues to real applications. Moreover, several other tuning methods have been proposed, including mechanically rotating the metamaterial [37], designing Kirigami structures [38] and electrically controlling the liquid crystal [39], etc. However, the development of devices with multifunctional polarization manipulation capabilities and with relatively easy applicability and a simple structure remains challenging.

In this paper, a new strategy of combining a metamaterial structure with our previously developed pneumatic actuator [40,41,42,43,44] is proposed. It can achieve multifunctional polarization manipulation at the same THz frequency range by dynamically controlling the applied air pressure. In the current case, the geometry of the specially designed metal spirals sitting on a suspended elastic membrane is changed under pneumatic actuation, affecting its interaction with the EM wave. From the simulation results, three types of polarization manipulation functionalities can be obtained at the same frequency band, from 1.3 THz to 1.5 THz, based on our proposed metamaterial. At the initial status without pneumatic actuation, polarization conversion from LP to CP can be obtained between the incidence and the reflection and vice versa. When it is actuated with the applied pressure of 100 kPa, a helicity converter can be achieved, in which the incident CP wave is kept as CP after reflection, whilst reversing its handedness. In contrast, if the applied pressure is changed to −50 kPa, the handedness of the reflected CP wave remains identical to that of the incident one, acting as a helicity keeper instead. Furthermore, the intrinsic mechanism of polarization manipulation is also discussed through analyzing the electric field distribution in the metamaterial during actuation.

## 2. Structure Design

The schematic of the proposed pneumatically actuated metamaterial (PAMM) is provided in Figure 1. Its unit cell is composed of two stacked components. The upper layer is a patterned 500 nm Au structure sitting on a suspended polydimethylsiloxane (PDMS) membrane with 4 μm thickness. The PDMS is used mainly due to its excellent elasticity and low absorption in the terahertz frequency range. In the current case, two Au Archimedean spirals with a 180° rotation angle along the clockwise direction are designed, the detailed parameters of which can be found in the attached table. The lower layer consists of a PDMS substrate with a cylindrical cavity being located at the center. The radius and the depth of the cylindrical cavity are designed to be 70 μm and 60 μm, respectively. The cavity is sealed by a suspended PDMS membrane using the oxygen plasma-assisted bonding method, forming the pneumatic actuator. At the same time, a uniform 500 nm Au film is deposited onto the cavity bottom surface. All the cavities in the periodically arranged unit cells are connected together via microchannels arranged at the cavity edges, providing access to the external pressure control system. Upon operation, with the introduction of air pressure into the cavities, the synchronous actuation of all the unit cells can be obtained, resulting in dynamic polarization modulation for the metamaterial.

Based on the proposed PAMM, three types of polarization manipulation functions can be obtained between the incidence and the reflection, as shown in Figure 2. At the initial “OFF” state without actuation, the PAMM acts as a reflective quarter-wave plate to achieve mutual conversion between LP and CP. For the “ON-1” state with 100 kPa actuation pressure, the PDMS membrane together with the attached Au structure will be deformed upward and the resultant PAMM will provide helicity conversion for the CP wave, similar to mirror reflection. In comparison, for the case of the “ON-2” state with downward deformation associated with applied pressure of −50 kPa, the handedness of the reflected CP wave will remain identical to that of the incident one, functioning as a helicity keeper instead.

## 3. Simulation Results

To study the polarization manipulation property, a simulation using the commercial software COMSOL Multiphysics 6.0 is performed, during which two types of simulation, namely mechanical simulation and EM simulation, are used. Firstly, the mechanical simulation is carried out to analyze the structure deformation under pneumatic actuation. As shown in Figure 3a, the circular PDMS membrane and the attached Au spirals are built in the model. The circumference of the PDMS membrane is set to be a clamped boundary and uniform pressure is applied onto the membrane surface, mimicking pneumatic actuation. The Young’s modulus and the Poisson’s ratio of the PDMS and Au are set to be 750 kPa/70 GPa and 0.49/0.44, respectively. Figure 3b shows the simulation results of the center membrane deformation under applied pressure from −50 kPa to 100 kPa. It can be clearly seen that the maximum upward and downward deflections at the membrane center can reach 47 μm and −44 μm, respectively. For better visualization, the deformed structures are also provided. Obviously, with the membrane’s deformation, the initial planar Au spirals will be transformed into 3D helixes with different handedness, and the corresponding helix parameters are determined by both the deformation direction and amplitude. For example, with the increasing positive pressure to 100 kPa, the right-handed Au helix will be obtained and the helix height will be gradually increased to 40 μm. In comparison, when negative pressure is applied, the left-handed Au helix will be generated instead, with an increase of its helix height to around 38 μm under −50 kPa. This structure transformation is the origin of the dynamic polarization manipulation.

Subsequently, the optical characteristics of the PAMM under different actuation statuses are also analyzed using the EM module in COMSOL Multiphysics. In the current case, the simulation model of the unit cell shown in Figure 4 is used, in which the open boundary condition is set in the z direction and the periodic boundary is set in both the x and y directions. The Au is modeled as a lossy metal with electric conductivity σ = 4.56 × 10^7^ S/m. The PDMS membrane is modeled as a dielectric material with a dielectric constant of 2.35 and loss tangent of 0.02. According to the skin effect concern, the metallic ground plane with 500 nm thickness can completely block all the incident EM waves to be transmitted. In order to disclose the resultant optical property change associated with pneumatic actuation, the geometry structures of the PDMS membrane and the Au spirals are refreshed with the deformed ones from the mechanical simulation and the EM simulation is performed again. Based on the combination of the mechanical and EM simulations, the effect of the pneumatic actuation can be analyzed and the polarization modulation performance can be obtained as well.

(1)“OFF” state

From EM theory, the electric field of any EM wave can be expressed as the vector synthesis of two orthogonal components, and Jones introduced a two-element column vector to describe the polarized wave (see Equation (1)).
(1)$$E=\left(\begin{array}{l} E_{x}\\ E_{y} \end{array}\right)=\left(\begin{array}{l} E_{ox}\exp\left(j\delta_{x}\right)\\ E_{oy}\exp\left(j\delta_{y}\right) \end{array}\right)=\frac{E_{ox}}{\sqrt{E_{ox}^{2}+E_{oy}^{2}}}\left(\begin{array}{l} \exp\left(j\delta_{x}\right)\\ a\exp\left(j\delta_{y}\right) \end{array}\right)\begin{array}{cc} & a=E_{oy}/E_{ox} \end{array}$$
where *E* and *δ* are the electric field and its phase, respectively.

For analysis purposes, the coordinate system shown in Figure 5 is used, in which the u-v coordinate system is obtained by rotating the x-y coordinate system 45 degrees counterclockwise. In this case, the relationship between the incident and the reflected electric fields can be expressed as follows [45]:(2)$$\left(\begin{array}{l} E_{u}^{r}\\ E_{v}^{r} \end{array}\right)=\left(\begin{array}{l} \begin{array}{cc} R_{uu} & R_{uv} \end{array}\\ \begin{array}{cc} R_{vu} & R_{vv} \end{array} \end{array}\right)\left(\begin{array}{l} E_{u}^{i}\\ E_{v}^{i} \end{array}\right)$$
where $E_{u}^{i}$ and $E_{v}^{i}$ denote the electric field magnitudes of the incident wave components in the u- and v-directions; $E_{u}^{r}$ and $E_{v}^{r}$ indicate those of the reflected wave components in the u- and v-directions, respectively. *R_uu_* and *R_vv_* represent the co-polarized components in the reflection, whilst *R_uv_* and *R_vu_* describe the cases of cross-polarization. The corresponding parameters of the reflection matrix R can be described as
$$R_{uu}=\left|R_{uu}\right|\exp\left(j\delta_{uu}\right),\;R_{uv}=\left|R_{uv}\right|\exp\left(j\delta_{uv}\right),\;R_{vu}=\left|R_{vu}\right|\exp\left(j\delta_{vu}\right),\;R_{vv}=\left|R_{vv}\right|\exp\left(j\delta_{vv}\right)$$

As for the “OFF” state, when the incident wave is linearly polarized along the y-direction, its corresponding Jones vector in the u-v coordinate system can be expressed as ${\left(\begin{array}{cc} E_{u}^{i} & E_{v}^{i} \end{array}\right)}^{T}={\left(\begin{array}{cc} 1 & 1 \end{array}\right)}^{T}/\sqrt{2}$. As a result, from Equation (2), the Jones vector of the reflected wave can be calculated as
(3)$$\left(\begin{array}{l} E_{u}^{r}\\ E_{v}^{r} \end{array}\right)=\left(\begin{array}{l} \frac{1}{\sqrt{2}}\left|R_{uu}\right|\exp\left(j\delta_{uu}\right)+\frac{1}{\sqrt{2}}\left|R_{uv}\right|\exp\left(j\delta_{uv}\right)\\ \frac{1}{\sqrt{2}}\left|R_{vu}\right|\exp\left(j\delta_{vu}\right)+\frac{1}{\sqrt{2}}\left|R_{vv}\right|\exp\left(j\delta_{vv}\right) \end{array}\right)$$

During analysis, the values of the involved parameters can be directly extracted from the EM simulation using COMSOL, in which the polarization direction of the incidence can be defined according to the u-v coordinate system. At the same time, the amplitudes of the reflection components along the u- and v-directions both can be extracted from the simulation by proper setting in the post-processing module in the software. For example, for the incident linear polarization along the u-direction, the obtained reflection amplitudes along the u- and v-directions are defined as *R_uu_* and *R_uv_*, respectively. Similarly, if the incidence is linearly polarized along the v-direction, the obtained reflection amplitudes along the u- and v-directions are, respectively, defined as *R_vu_* and *R_vv_* instead.

From the results shown in Figure 6, it can be seen that within the studied frequency range from 1.3 THz to 1.5 THz, $\left|R_{uu}\right|\approx \left|R_{vv}\right|$, $\left|R_{uv}\right|\approx \left|R_{vu}\right|$, $\delta_{uv}\approx \delta_{vu}$ and $\delta_{uu}+90^{\circ }\approx \delta_{vv}$. Considering the relatively low intensity of the cross-polarized reflection components $\left|R_{uv}\right|$ and $\left|R_{vu}\right|$, they can be ignored in the following analysis. As a result, the Jones vector of the reflection can be rewritten as
(4)$$\left(\begin{array}{l} E_{u}^{r}\\ E_{v}^{r} \end{array}\right)=\left(\begin{array}{l} \frac{1}{\sqrt{2}}\left|R_{uu}\right|\exp\left(j\delta_{uu}\right)\\ \frac{1}{\sqrt{2}}\left|R_{uu}\right|\exp\left(j\left(\delta_{uu}+90^{\circ }\right)\right) \end{array}\right)=\left(\begin{array}{l} \frac{1}{\sqrt{2}}\left|R_{uu}\right|\exp\left(j\delta_{uu}\right)\\ \frac{1}{\sqrt{2}}\left|R_{uu}\right|\exp\left(j\delta_{uu}\right)\cdot j \end{array}\right)$$

Obviously, $E_{u}^{r}\cdot j=E_{v}^{r}$. Given this condition and considering the opposite transmission direction of the reflection to that of the incidence, the reflection can be deduced to be an RCP wave.

**Figure 6 micromachines-14-02094-f006:**
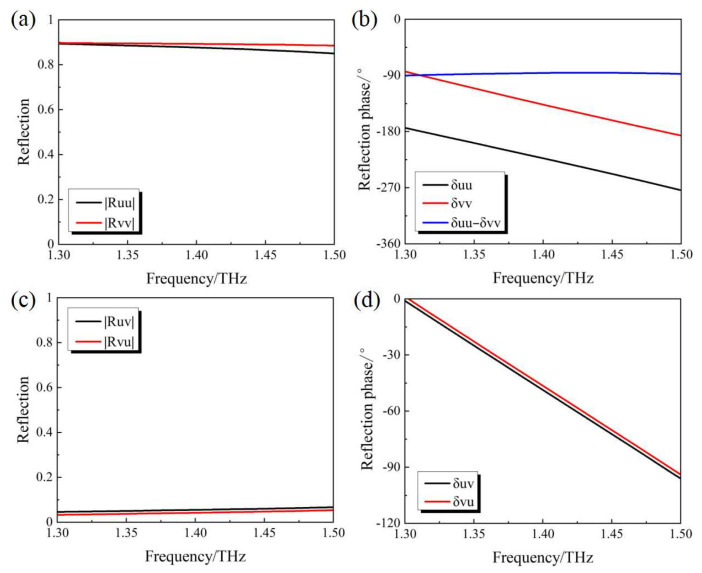
Simulation results of (**a**) $\left|R_{uu}\right|$ and $\left|R_{vv}\right|$ (**b**) $\delta_{uu}$ and $\delta_{vv}$ (**c**) $\left|R_{uv}\right|$ and $\left|R_{vu}\right|$ (**d**) $\delta_{uv}$ and $\delta_{vu}$ for the “OFF” state of the PAMM.

In the case of the incidence with an RCP wave, its corresponding Jones vector can be expressed as ${\left(\begin{array}{cc} E_{u}^{i} & E_{v}^{i} \end{array}\right)}^{T}={\left(1-j\right)}^{T}/\sqrt{2}$. From the above analysis, the Jones vector of the current reflected wave can be deduced to be
(5)$$\left(\begin{array}{l} E_{u}^{r}\\ E_{v}^{r} \end{array}\right)=\left(\begin{array}{l} \frac{1}{\sqrt{2}}\left|R_{uu}\right|\exp\left(j\delta_{uu}\right)\\ \frac{1}{\sqrt{2}}\left|R_{uu}\right|\exp\left(j\left(\delta_{uu}+90^{\circ }\right)\right)\cdot \left(-j\right) \end{array}\right)=\left(\begin{array}{l} \frac{1}{\sqrt{2}}\left|R_{uu}\right|\exp\left(j\delta_{uu}\right)\\ \frac{1}{\sqrt{2}}\left|R_{uu}\right|\exp\left(j\delta_{uu}\right) \end{array}\right)$$

Given the condition of $E_{u}^{r}=E_{v}^{r}$, the reflection with linear polarization along the y-direction can be obtained. It is obvious that the mutual conversion between the LP wave along the y-direction and the RCP wave can be obtained after being reflected. Similarly, the mutual polarization conversion between the incident LP wave along the x-direction and the LCP wave can also be validated.

(2)“ON-1” and “ON-2” state

Further study of the polarization manipulation characteristics of the PAMM under pneumatic actuation is carried out. Figure 7a,b show the simulation results of the R−+ and R++ with the applied pressure changing from −50 kPa to 100 kPa. R−+ and R++ represent the intensity of the LCP and RCP components in the reflection under RCP incidence. It can be seen that with the increasing applied pressure from −50 kPa to 100 kPa, the reflection R−+ increases from 0 to above 0.8, whilst the reflection R++ decreases from 0.9 to around 0.1. As for the “ON-1” state with the applied pressure of 100 kPa, much larger R−+ can be obtained when compared with R++, which indicates that the RCP incident can be almost converted into LCP after being reflected. In contrast, in the case of the “ON-2” state with the applied pressure of −50 kPa, nearly zero R−+ and R++ higher than 0.9 can be found, demonstrating good handedness maintenance between the incidence and the reflection. A similar polarization conversion capability for the case of LCP incidence (namely polarization conversion from LCP to RCP (R+−) under “ON-1” state and handedness maintenance (R−−)under “ON-2” state) can also be obtained according to the simulation results shown in Figure 7c,d. Moreover, when no pressure is applied, which corresponds to the above-mentioned “OFF” state, nearly equal values of R−+ and R++ (also the R+− and R−−) can be found. Considering the fact that the LP wave can be synthesized by the RCP wave and the LCP wave with equal amplitudes, the polarization conversion from CP to LP under this state can be validated as well, agreeing well with the above result.

Similarly, the Jones matrix analysis method as mentioned above is also used to study the polarization conversion under the “ON-1” and “ON-2” states. The relationship between the incident and the reflected electric fields in the x-y coordinate system can be expressed as follows:(6)$$\left(\begin{array}{l} E_{x}^{r}\\ E_{y}^{r} \end{array}\right)=\left(\begin{array}{l} \begin{array}{cc} R_{xx} & R_{xy} \end{array}\\ \begin{array}{cc} R_{yx} & R_{yy} \end{array} \end{array}\right)\left(\begin{array}{l} E_{x}^{i}\\ E_{y}^{i} \end{array}\right)$$

As for the “ON-1” state with RCP incidence, when substituting its Jones vector in the x-y coordinate system, being described as ${\left(1-j\right)}^{T}/\sqrt{2}$, into Equation (6), the Jones vector of the reflection can be calculated as
(7)$$\left(\begin{array}{l} E_{x}^{r}\\ E_{y}^{r} \end{array}\right)=\left(\begin{array}{l} \frac{1}{\sqrt{2}}\left|R_{xx}\right|\exp\left(j\delta_{xx}\right)+\frac{1}{\sqrt{2}}\left|R_{xy}\right|\exp\left(j\delta_{xy}\right)\cdot \left(-j\right)\\ \frac{1}{\sqrt{2}}\left|R_{yx}\right|\exp\left(j\delta_{yx}\right)+\frac{1}{\sqrt{2}}\left|R_{yy}\right|\exp\left(j\delta_{yy}\right)\cdot \left(-j\right) \end{array}\right)$$

Figure 8 shows the values of the involved parameters extracted from the EM simulation. It can be seen that $\left|R_{xx}\right|\approx \left|R_{yy}\right|$, $\left|R_{xy}\right|\approx \left|R_{yx}\right|$, $\delta_{xy}\approx \delta_{yx}$ and $\delta_{xx}\approx \delta_{yy}$. Considering the relatively low intensity of the cross-polarized reflection components $\left|R_{xy}\right|$ and $\left|R_{yx}\right|$, they will be ignored in the following analysis.

Combining the above simulation results with Equation (6), the Jones vector of the reflection can be obtained:(8)$$\left(\begin{array}{l} E_{x}^{r}\\ E_{y}^{r} \end{array}\right)=\left(\begin{array}{l} \frac{1}{\sqrt{2}}\left|R_{xx}\right|\exp\left(j\delta_{xx}\right)\\ \frac{1}{\sqrt{2}}\left|R_{xx}\right|\exp\left(j\delta_{xx}\right)\cdot \left(-j\right) \end{array}\right)$$It can be found in Equation (8) that $E_{x}^{r}\cdot \left(-j\right)=E_{y}^{r}$. Compared with the Jones vector of the RCP incidence and considering the reversed transmission direction, LCP reflection can be concluded, thus providing the polarization conversion from RCP incidence to LCP reflection under the “ON-1” state. Similarly, the polarization conversion from LCP to RCP can also be verified between the incidence and the reflection.

The same analysis method is also used for the “ON-2” state. Through substituting the extracted parameters from the electromagnetic simulation for the case of RCP incidence as shown in Figure 9 into Equation (7) (where $\left|R_{xx}\right|\approx \left|R_{yy}\right|$, $\left|R_{xy}\right|\approx \left|R_{yx}\right|$, $\delta_{xy}\approx \delta_{yx}$ and $\delta_{xx}+180^{\circ }\approx \delta_{yy}$), the Jones vector of the reflection can be calculated as
(9)$$\left(\begin{array}{l} E_{x}^{r}\\ E_{y}^{r} \end{array}\right)=\left(\begin{array}{l} \frac{1}{\sqrt{2}}\left|R_{xx}\right|\exp\left(j\delta_{xx}\right)+\frac{1}{\sqrt{2}}\left|R_{xy}\right|\exp\left(j\delta_{xy}\right)\cdot \left(-j\right)\\ \frac{1}{\sqrt{2}}\left|R_{xy}\right|\exp\left(j\delta_{xy}\right)+\frac{1}{\sqrt{2}}\left|R_{xx}\right|\exp\left(j\cdot \delta_{xx}\right)\cdot \left(-1\right)\cdot \left(-j\right) \end{array}\right)$$

From the result of $E_{x}^{r}\cdot j=E_{y}^{r}$, RCP reflection can be obtained in the current status, demonstrating the same polarization status as that of the incidence. The same polarization characteristic can also be found in the case of LCP incidence, disclosing the stable polarization maintenance operation of the proposed PAMM under the “ON-2” state.

All the polarization modulation functions mentioned above based on the proposed PAMM are summarized in the Table 1. It is obvious that different polarization manipulation capabilities can be achieved with the same device based on the switching of the operation states using pneumatic actuation.

## 4. Discussion

In order to reveal the intrinsic polarization manipulation mechanism associated with the PAMM, its frequency responses regarding absorption under different operation statuses are studied, as shown in Figure 10, in which the X mode and Y mode represent the linearly polarized incidence along the x- and y-directions, respectively. In the case of the X mode, shown in Figure 10a, the absorption is calculated by $A=1-\left|R_{xx}\right|-\left|R_{yx}\right|$. In comparison, for the Y mode, shown in Figure 10b, the absorption is calculated by $A=1-\left|R_{yy}\right|-\left|R_{xy}\right|$.

From Figure 10, it can be seen that the PAMM exhibits a distinct absorption peak at the same frequency within the frequency range from 1.3 THz to 1.85 THz for both the X and Y modes, and an obvious peak shift as well as amplitude variation can also be found associated with the change in the pneumatic actuation pressure. With the decreasing actuation pressure from 100 kPa to 0 kPa, the resonant peak will be moved towards a higher frequency. Moreover, for the X mode operation, a monotonous decrease in the resonant amplitude will be induced. In comparison, the amplitude will be increased first and then decreased for the case of the Y mode. With the further decrease in the actuation pressure to −50 kPa, the resonant peaks will be shifted outside of the interested frequency band. Due to the discrepancy between the resonance change tendencies in the X and Y modes during actuation, the phase difference between the reflections of different components (namely *δ_xx_*-*δ_yy_*) will be changed. As shown in Figure 11, the phase difference at 1.5 THz is selected for demonstration, in which it is increased from nearly −180° to 0° with increasing actuation pressure from −50 kPa to 100 kPa. It is obvious that with the change in the phase difference, the Jones vector of the structure will be changed accordingly, thus modulating the polarization characteristic.

For further analysis, the current distributions in the Au spiral structures at the absorption peak of 1.59 THz under the actuation pressure of 100 kPa are specifically extracted, as shown in Figure 12. It can be seen that there exists obvious current flow in the Au spiral structures in both of the two operation modes, and the flow directions within different regions are also quite different. Considering the Ampere rule, the magnetic field will be generated associated with the current flow, and the spatial magnetic fields from different regions will interact with each other. It is thought to be this magnetic coupling effect that causes the absorption peak.

The corresponding electric field (E-field) distributions in the device’s top layer at individual absorption peaks are provided in Figure 13. It can be seen that most electric fields are mainly concentrated within the center gap region of the Au spirals in all cases, representing strong coupling between Au spirals. In the case of the X operation mode, with the change in the applied pneumatic actuation pressure from 100 kPa to 0 kPa, a monotonous reduction in the E-field strength can be found, agreeing well with the tendency of the gradually decreasing absorption peak amplitude. A similar relationship can also be obtained for the device operating under the Y mode, in which the E-field strength as well as the absorption peak amplitude will be firstly increased when the applied pressure is decreased from 100 kPa to 30 kPa. Subsequently, with the further decrease in the applied pressure, both of them will be continuously reduced. Obviously, the amplitude of the absorption peak is dominated by the coupling condition of the PAMM, which is dependent on its geometric profile.

Moreover, combining the structure configuration with the device operation, a Fabry–Perot (F-P) cavity-like structure will be generated between the top Au spirals and the ground Au layer. With the decrease in the applied pressure from 100 kPa to 0 kPa, the resultant structure deformation will also be reduced, thus reducing the gap between the top Au spirals and the ground Au layer, as well as the F-P cavity length. Considering the fact that, within the free spectral range, the resonant frequency of the F-P cavity is inversely proportional to its cavity length, its resonant peak should be shifted to a higher frequency, which agrees well with the above simulation results.

## 5. Conclusions

In summary, a new strategy is proposed to achieve multifunctional polarization conversion at THz frequencies with the same device by integrating a pneumatic actuator into a specifically designed reflective-type metamaterial. During operation, through dynamic control of the applied actuation pressure, the geometries of the metamaterial will be mechanically changed, affecting its interaction with the incident EM waves. From the simulation and theoretical analysis, on-demand switching among three types of polarization manipulation capabilities has been successfully demonstrated. With respect to the initial “OFF” state without pneumatic actuation, the PAMM functions as a reflective quarter-wave plate to provide mutual conversion between LP and CP. In comparison, for the “ON-1” state with applied actuation pressure of 100 kPa, the PAMM can achieve polarization conversion from RCP to LCP and vice versa, acting as mirror reflection. When the applied pressure is reduced to −50 kPa, another operation state called “ON-2” will result, in which the handedness of the reflected CP wave will remain identical to that of the incident one instead. Moreover, the absorption spectra of the PAMM under different statuses have been studied and the observed absorption peak change with respect to the actuation pressure has also been discussed by analyzing the corresponding electric field distribution as well as the equivalent F-P cavity structure, which is thought to be the origin of the polarization manipulation. Considering the compact integration, easy control and flexible function associated with the current strategy, it offers a promising candidate for the building of new, functional THz devices for various applications.

## Figures and Tables

**Figure 1 micromachines-14-02094-f001:**
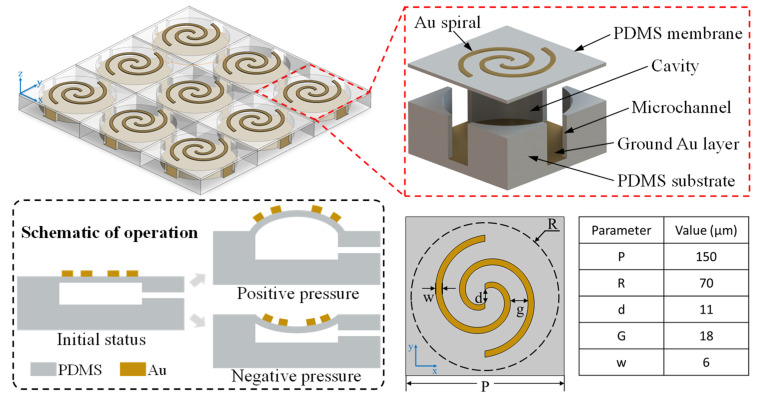
Schematic of the proposed pneumatic actuated metamaterial.

**Figure 2 micromachines-14-02094-f002:**
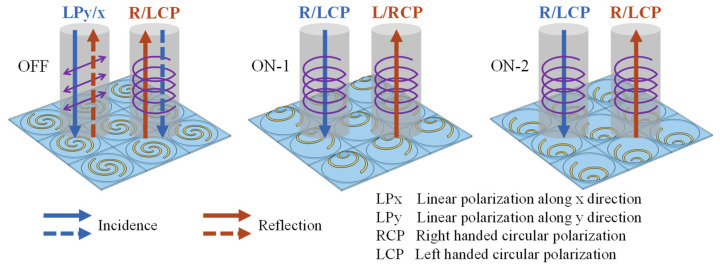
Polarization manipulation capabilities corresponding to three states of the PAMM.

**Figure 3 micromachines-14-02094-f003:**
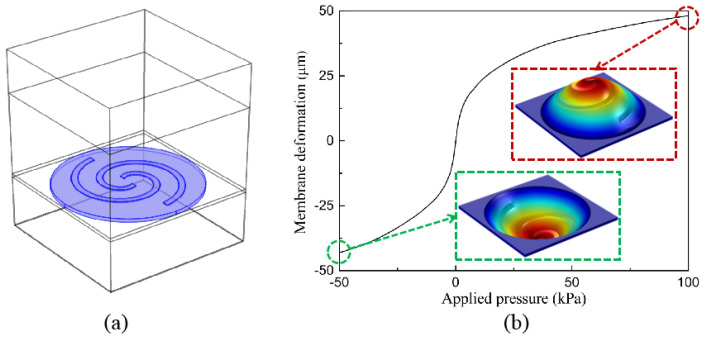
Mechanical simulation for the PAMM. (**a**) Schematic of the model used in mechanical simulation. (**b**) Simulation results of the membrane deformation as a function of applied pressure.

**Figure 4 micromachines-14-02094-f004:**
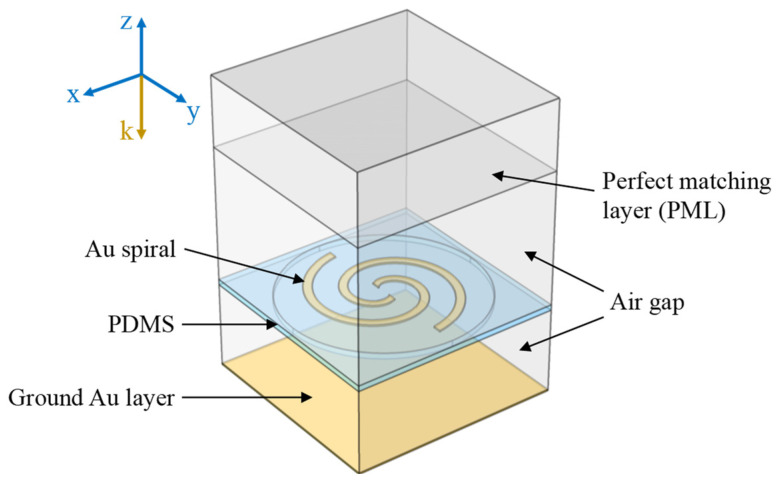
Schematic of the model used in electromagnetic simulation.

**Figure 5 micromachines-14-02094-f005:**
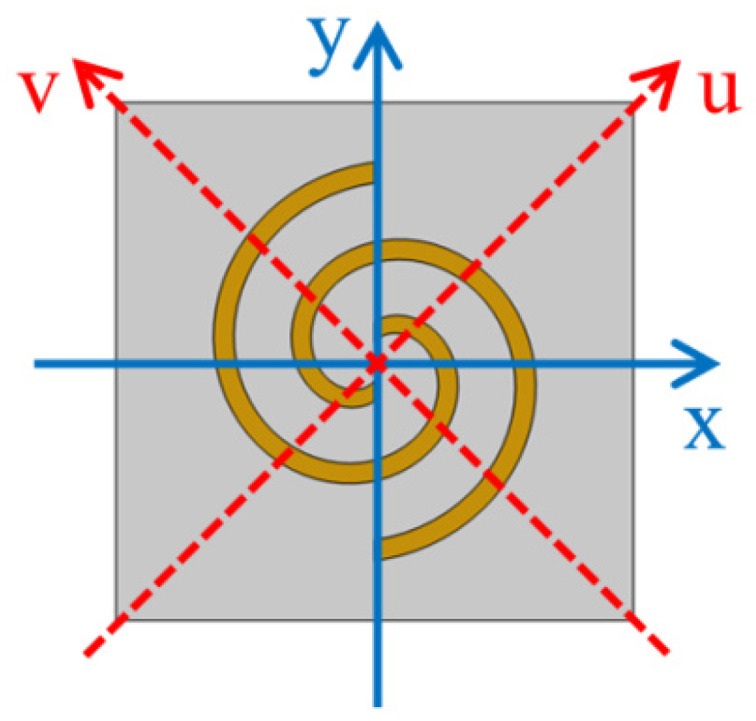
Schematic of the x-y and u-v coordinate system.

**Figure 7 micromachines-14-02094-f007:**
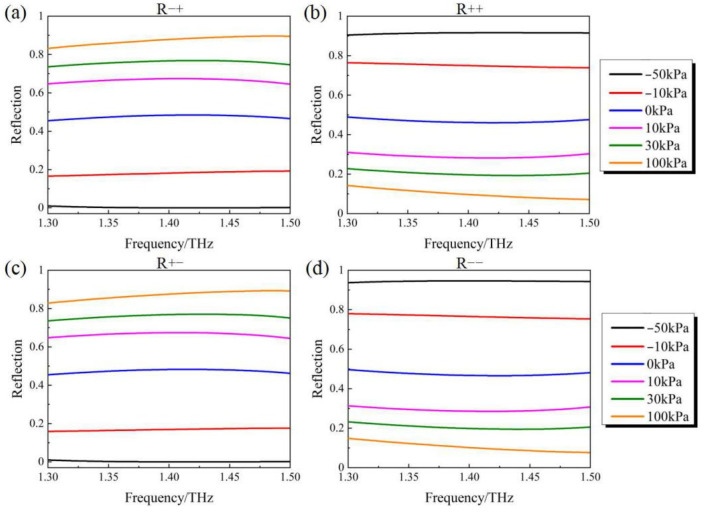
Simulation results of (**a**) R−+ (**b**) R++ (**c**)R+− (**d**)R−− under pneumatic actuation with different pressures.

**Figure 8 micromachines-14-02094-f008:**
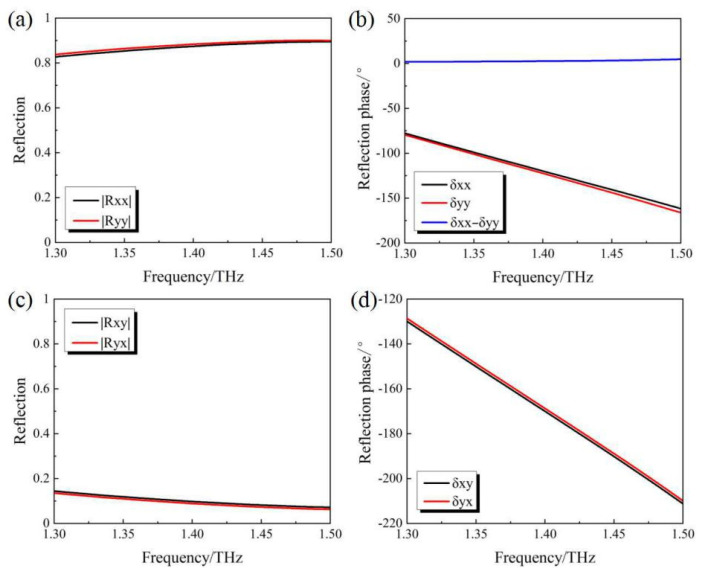
Simulation results of (**a**) $\left|R_{xx}\right|$ and $\left|R_{yy}\right|$ (**b**) $\delta_{xx}$ and $\delta_{yy}$ (**c**) $\left|R_{xy}\right|$ and $\left|R_{yx}\right|$ (**d**) $\delta_{xy}$ and $\delta_{yx}$ for the “ON-1” state of the PAMM.

**Figure 9 micromachines-14-02094-f009:**
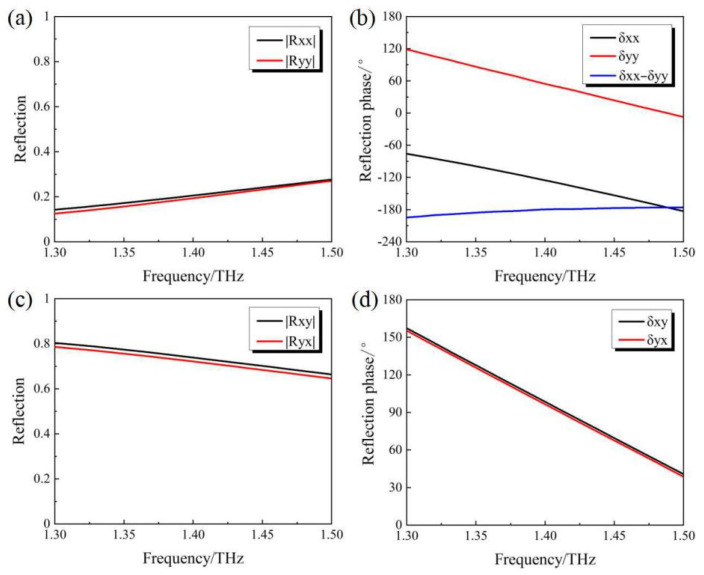
Simulation results of (**a**) $\left|R_{xx}\right|$ and $\left|R_{yy}\right|$ (**b**) $\delta_{xx}$ and $\delta_{yy}$ (**c**) $\left|R_{xy}\right|$ and $\left|R_{yx}\right|$ (**d**) $\delta_{xy}$ and $\delta_{yx}$ for the “ON-2” state of the PAMM.

**Figure 10 micromachines-14-02094-f010:**
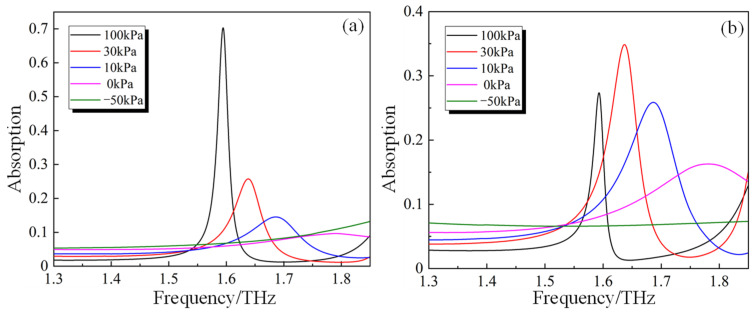
Simulation results of the absorption under pneumatic actuation with different pressures of (**a**) X mode and (**b**) Y mode.

**Figure 11 micromachines-14-02094-f011:**
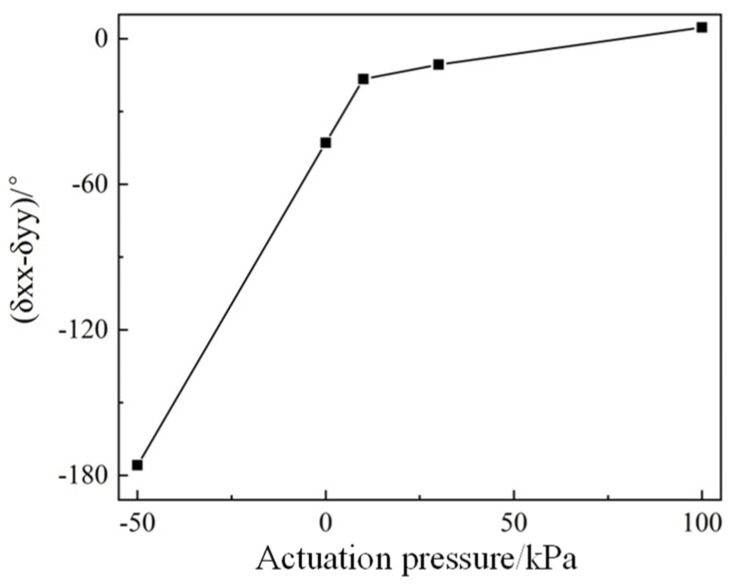
Simulation result of the phase difference *δ_xx_*-*δ_yy_* as a function of the actuation pressure.

**Figure 12 micromachines-14-02094-f012:**
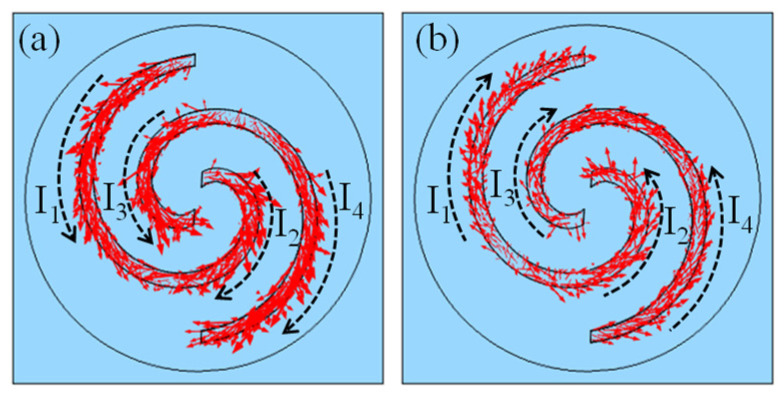
Simulation results of the current distribution in the Au spirals of (**a**) X mode and (**b**) Y mode.

**Figure 13 micromachines-14-02094-f013:**
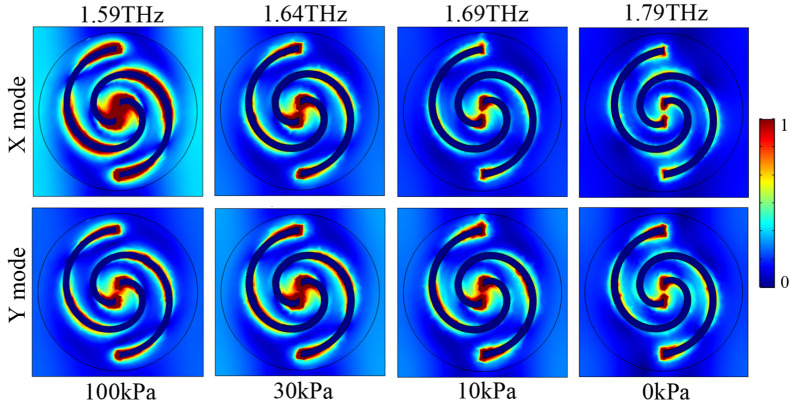
Simulation results of the electric field distributions in the device top layer under different operation conditions.

**Table 1 micromachines-14-02094-t001:** Polarization manipulation capabilities corresponding to three states of the PAMM.

State	Incidence	Reflection
OFF	LPy	RCP
	LPx	LCP
	RCP	LPy
	LCP	LPx
ON-1	RCP	LCP
	LCP	RCP
ON-2	RCP	RCP
	LCP	LCP

## Data Availability

Data are contained within the article.

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
