# Peer review of "On-Demand Dynamic Terahertz Polarization Manipulation Based on Pneumatically Actuated Metamaterial"

_micromachines, 2023, doi:10.3390/mi14112094_

Round 1
Reviewer 1 Report
Comments and Suggestions for Authors
This paper proposes a novel metamaterial design that can achieve dynamic polarization manipulation in the terahertz frequency range through pneumatic actuation. The metamaterial consists of gold spiral structures on a PDMS membrane suspended above a cavity. By applying positive or negative pressure to the cavity, the membrane deforms, changing the geometry of the gold spirals and modifying their interaction with the incident electromagnetic waves. The investigation are interesting, but some critical factors are not clear. Thus, authors have to make necessary modifications for this manuscript.
1. The outdated references in the introduction are a major weakness of this paper. The lack of recent, relevant citations significantly undermines the contextual framing and suggests the authors may not have a comprehensive grasp of the current state of the field. While the proposed metamaterial design shows promise, failing to properly cite the most advanced prior work severely limits the ability to evaluate the true novelty and contributions of this research. Without citations to key recent studies on terahertz metamaterials and reconfigurable/actuatable designs, it is difficult to determine how much of an advance this pneumatic metamaterial presents. The significance and originality of the work are unclear. Given how rapidly research in terahertz metamaterials and devices has progressed in the past decade, the introduction as written casts serious doubt regarding the authors' understanding of the contemporary literature. This omission must be addressed through citing the latest highly-relevant papers on tunable terahertz metamaterials and polarization control.
2. The authors do not provide sufficient analysis specifically tied to the Achimedean spirals of the deformed gold spirals under pneumatic actuation. The author should expand the Jones matrix analysis to connect the observed polarization behavior more conclusively to the geometric changes in the Achimedean spirals for the 3 pressure conditions. Simulate the current distributions in the double helix resonators under actuation to reveal the resonance mechanisms leading to polarization conversion vs. maintenance.
3. Provide more discussion and ideally simulations of how the handedness and pitch of the induced helices under pressure influence the polarization effects.
4. Expand a bit more on comparisons with other recent work in active/reconfigurable THz metamaterials.
5. I suggest authors revise fig.1, and as ref. [16]. The current fig. 1 may make misunderstanding to reader as the up layer and low layer are separated.
6. Some grammar mistakes should be revised.
Reviewer 2 Report
Comments and Suggestions for Authors
A new way to get different polarizations in reflected wave combining metamaterial structure with pneumatic actuator is proposed. It can achieve three polarizations, namely linear and two circularly polarized (CP) with different directions of rotation at the same THz frequency range. The paper is completely theoretical and simulation using commercial software COMSOL Multiphysics is performed. The circular PDMS membrane and the attached Au spirals are built in the model. The radius and the depth of the cylindrical cavity are designed to be 70 μm and 60 μm. The cavity is sealed by the suspended PDMS membrane, forming the pneumatic actuator. During operation, through dynamic control of the applied actuation pressure, the geometries of the spiral will be pneumatically changed, affecting its interaction with the incident linearly polarized electromagnetic waves. From simulation results, three types of polarization manipulation functionality can be obtained at the same frequency band from 1.3 THz to 1.5 THz. When the device is actuated with the applied pressure of 100 kPa, a helicity converter will be achieved, in which the incident CP wave will still be kept as CP after reflection, whilst reversing its handedness. In contrary, if the applied pressure is changed to -50 kPa, the handedness of the reflected CP wave will remain identical to that of the incident one.
The paper falls into the scope of the journal. Conclusion of the paper is mostly supported by the text. The list of references is appropriate. The paper can be published after some corrections will be introduced.
1. The practical application of the proposed air pressure actuator causes a question how reliable this actuator is expected to be? Can the authors estimate how many cycles of switching this actuator can to fulfill before seal failure or any other damage?
2. Several misprints have to be corrected in lines 84, 104, 132, 134,154.
3. Using of the bold type is not required in lines 160,161, 162, 165, 168, 174, 217, 222, 239,260,261 and formulas (3), (4), (6), (7), (8).
Reviewer 3 Report
Comments and Suggestions for Authors
Manuscript ID_micromachines-2696135
Review for “On-demand dynamic terahertz polarization manipulation based on pneumatically actuaded metamaterial”
The authors present interesting information on a new tuning strategy for metamaterials: the inclusion of pneumatically-actuated elements that achieve on-demand THz polarization control. The article is within the scope of the journal and presents enough original elements to be taken into consideration. I can recommend publication after the following observations are addressed:
· The introduction can be further enhanced with the following new research: https://doi.org/10.3390/s21165600 and https://doi.org/10.3390/polym13111860, as well as their references. The papers directly tackle tunability as a function of actuated metasurface elements.
· It is unclear how the actuation works. A schematic view of the actuation principle would provide additional information on the matter.
· The authors have to specify the effect on reflection that the deformation of the membrane has, by providing information on the interface flatness upon deformation.
· It is unclear whether an absorption effect in the metasurface is desired for the polarization controller. The authors should add a paragraph in which they discuss advantages/disadvantages to this behavior.
· The conductivity of Au is written wrongly (line 134). Please correct.
· It is unclear if the values of R_uu , R_vv, R_uv, R_vu are extracted from theory or from the simulation, and how. Please specify.
Under current circumstances, I recommend a Major Revision in which the following observations are addressed.
Round 2
Reviewer 1 Report
Comments and Suggestions for Authors
The manuscript is revised as comments. I consider this paper satisfy the journal's requirement and recommend it published as current form.
Reviewer 2 Report
Comments and Suggestions for Authors
The paper can be published in the presented view.
Reviewer 3 Report
Comments and Suggestions for Authors
Manuscript ID_micromachines-2696135
Review for “On-demand dynamic terahertz polarization manipulation based on pneumatically actuaded metamaterial”
The authors present interesting information on a new tuning strategy for metamaterials: the inclusion of pneumatically-actuated elements that achieve on-demand THz polarization control. The article is within the scope of the journal and presents enough original elements to be taken into consideration. I can recommend publication after the following observation is addressed:
The correct citation for reference [13] is : O. Dănilă, D. Mănăilă-Maximean, A. Bărar, and V.A. Loiko, V.A., 2021. Non-layered gold-silicon and all-silicon frequency-selective metasurfaces for potential mid-infrared sensing applications. Sensors, 21(16), p.5600.
Please correct.
Recommendation: Minor revision